GAC3D: improving monocular 3D object detection with ground-guide model and adaptive convolution

Bui Minh-Quan Viet 1 2
Ngo Duc Tuan 1 2
http://orcid.org/0000-0002-5806-5910 Pham Hoang-Anh 1 2
http://orcid.org/0000-0001-7321-7401 Nguyen Duc Dung 1 2 nddung@hcmut.edu.vn
1 Computer Science and Engineering Faculty, Ho Chi Minh City University of Technology (HCMUT) , Ho Chi Minh City , Vietnam
2 Vietnam National University Ho Chi Minh City (VNU-HCM) , Ho Chi Minh City , Vietnam
Shang Yilun
Electronic publication date: 2021 Oct 6
Publication date: 2021
Volume: 7
Electronic Location ID: e686
Received 2021 Apr 20; Accepted 2021 Aug 2
Copyright: © 2021 Bui et al.
Copyright year: 2021
Copyright holder: Bui et al.
License: This is an open access article distributed under the terms of the Creative Commons Attribution License, which permits unrestricted use, distribution, reproduction and adaptation in any medium and for any purpose provided that it is properly attributed. For attribution, the original author(s), title, publication source (PeerJ Computer Science) and either DOI or URL of the article must be cited.
License URL: https://creativecommons.org/licenses/by/4.0/

Keywords: 3D object detection, Monocular, Adaptive convolution, Ground-guide, Depth estimation, Pseudo-pose

Funding: Ho Chi Minh City University of Technology To-KHMT-2020-03 This research is supported by Ho Chi Minh City University of Technology (HCMUT), VNU-HCM under grant number To-KHMT-2020-03. The funders had no role in study design, data collection and analysis, decision to publish, or preparation of the manuscript.

==============================
Monocular 3D object detection has recently become prevalent in autonomous driving and navigation applications due to its cost-efficiency and easy-to-embed to existent vehicles. The most challenging task in monocular vision is to estimate a reliable object’s location cause of the lack of depth information in RGB images. Many methods tackle this ill-posed problem by directly regressing the object’s depth or take the depth map as a supplement input to enhance the model’s results. However, the performance relies heavily on the estimated depth map quality, which is bias to the training data. In this work, we propose depth-adaptive convolution to replace the traditional 2D convolution to deal with the divergent context of the image’s features. This lead to significant improvement in both training convergence and testing accuracy. Second, we propose a ground plane model that utilizes geometric constraints in the pose estimation process. With the new method, named GAC3D, we achieve better detection results. We demonstrate our approach on the KITTI 3D Object Detection benchmark, which outperforms existing monocular methods.

Introduction

In recent years, with the evolution of the deep neural network in computer vision (Krizhevsky, Sutskever & Hinton, 2012; He et al., 2016; Brock et al., 2021), we have seen various methods being proposed to resolve 2D object detection task (Ren et al., 2015; He et al., 2017; Redmon & Farhadi, 2018; Zhou, Wang & Krähenbühl, 2019; Tan, Pang & Le, 2020) and achieve remarkable performances, which almost approach human visual perception. Even so, in particular fields such as autonomous driving or infrastructure-less robot navigation, the demand for scene understanding, including the detailed 3D poses, identities, and scene context, is still high. Researchers pay attention to 3D object detection, especially in autonomous navigation applications. To obtain an accurate depth map of the environment, people adopt LiDAR sensors widely due to their reliable 3D point cloud acquired using laser technology. Such LiDAR-based systems (Shi, Wang & Li, 2019; Lang et al., 2019; Shi et al., 2020; He et al., 2020) achieve promising results, they also come with visible limitations, including high-cost sensors, hard to mount on vehicles, sparse and unstructured data. Therefore, an alternative solution using a singular RGB camera is required. It is far more affordable, versatile, and almost available on every modern vehicle. The main difficulty of image-based 3D object detection is the missing depth information, which results in a significant gap performance compared to LiDAR-based methods. While stereo systems are available, we have to calibrate the cameras with relatively high accuracy. Commercial cameras such as Bumblebee is a compact system with lower cost; however, the obtained depth map quality is nowhere near the standard required for autonomous driving systems, especially for outdoor. Estimating the depth via a single camera is a good choice since human also perceives the depth from 2D images. However, the accuracy of monocular depth estimation is not as good as ToF cameras like LiDAR. That makes the 3D object detection task on a single camera more challenging and has brought much attention from the community. In this work, we propose a 3D object detection system based on a single camera view. We also demonstrate that our proposed framework can bridge the gap between LiDAR and image-based detectors.

There are two main approaches in monocular 3D object detection: the representation transformation and the 2D convolutional-neural-network (CNN). In the representation approach, the general idea is to imitate the 3D point clouds of LiDAR by estimating depth information from images. This depth map is then projected into 3D space to generate the pseudo-point clouds. With the pseudo-point clouds, one can employ several algorithms that using LiDAR data to detect objects. However, the raw point clouds are sparse due to the range laser sensor’s physical principle. The pseudo-point clouds, however, are considerably denser and depend heavily on the estimated depth map quality. Thus, applying object pose detection on these pseudo-point clouds decreases the performance significantly. Besides, the pseudo-LiDAR methods consist of several separate steps, which usually require training the estimation model separately, leading to non-end-to-end optimal training and time-consuming in the inference phase.

On the other hand, the 2D CNN approaches extend the 2D object detector’s architecture to adapt the 3D output representation and add several techniques to solve the ill-posed problem. In M3D-RPN (Brazil & Liu, 2019), and D4LCN (Ding et al., 2020), the authors redefine the anchor from YOLO (Redmon & Farhadi, 2018) to 3D anchor by adding dimension, orientation, and depth information. Liu et al. (2019), Naiden et al. (2019) follow the pipeline of two-stage detector like Faster R-CNN (Ren et al., 2015) to detect the 2D bounding box of the object, then regress to the 3D box. In the proposal stage, they localize the 2D bounding boxes with additional 3D regressed attributes. In the second stage, the 3D bounding box can be reconstructed via an optimization process that leverage the geometric constraints between the projected 3D box and the corresponding 2D box. These methods rely heavily on an accurate 2D detector. Even with a small error in the 2D bounding box, it can cause poor 3D prediction.

Inspired by anchor-free architecture in the 2D detector CenterNet (Zhou, Wang & Krähenbühl, 2019), SMOKE (Liu, Wu & T’oth, 2020) and RTM3D (Li et al., 2020) add several regression heads parallel to the primary 2D center detection head to regress 3D properties. These anchor-free approaches are more light-weight and flexible than the other anchor-based approaches since they do not have to pre-define the 3D anchor box, which is more complicated than the one in the 2D detector. Unlike some regular 2D object detection datasets such as COCO (Lin et al., 2014) and Pascal VOC (Everingham et al., 2009), the 3D datasets like KITTI (Geiger, Lenz & Urtasun, 2012) usually contain occluded objects due to the driving scenario of the data collecting process. As illustrated in Fig. 1, in a dense object scene, the 2D center of the occluded car locates at another instance’s appearance, which potentially causes errors in the pose estimation process. Moreover, the standard convolution operation is content-agnostic (Su et al., 2019), which means once trained, the kernel weights remain unchanged despite the variance of the input scenario. Thus, the center misalignment phenomenon confounds the center-based detector with traditional convolution filtering to identify the object locations accurately. To overcome this issue, we introduce a novel convolution operation called depth adaptive convolution layer, which leverages the external guidance from a pre-trained depth estimator to enhance features selection for regression and detection tasks. Our novel convolution filtering applies a set of secondary weights on the original convolution kernel based on the depth value variance at a single pixel. As a consequence, this operator improves the precision and robustness of center-based object prediction.

Figure 1 An example of an occluded object in the driving scenario: the red dot representing the 2D center of the car lies on the visual appearance of the other car.

In autonomous navigation and robotics applications, most moving obstacles stand on a ground plane. Thus, the height difference between the mounted camera and the obstacle’s bottoms is almost constant for each vehicle and is equal to the camera’s height. Moreover, assume that the ground plane is parallel to the axis of the camera, we can reproject the 2D location in the image to the ground plan to get the z-coordinate of the object bottom. This assumption holds for most real driving scenarios. The reprojection can significantly reduce the lack of depth information in the monocular image. Such geometric information obtained from these perspective priors can help mitigate the ill-posed problem in monocular 3D object detection.

In this work, we proposed a single-stage monocular 3D object detector employing ideas from the above discussions. We name this method GAC3D (Geometric ground-guide and Adaptive Convolution for 3D object detection). Figure 2 provides an overview of our proposed detection framework. Our work consists of two main contributions:

Employ a novel depth adaptive convolution playing as secondary weights to adapt with the depth variance on every pixel.

Introduce a ground-guide module to infer the 3D object bounding box information from 2D regression results. For this approach, we introduce the concept of pseudo-position that serves as an initial value for the estimation process.

Figure 2 The overview of our proposed architecture for the monocular 3D object detection framework.

The backbone network extracts features from the RGB image. Then, we apply the depth adaptive convolution detection head to predict the object center, pseudo-contact point offsets, keypoints offsets, observation angle, and dimension. The Geometric Ground Guide Module takes the intermediate results from detection heads and the pseudo-position value to recover the object’s 3D pose via 2D–3D geometric transformation.

The details of our proposed framework will be presented in the next section. We demonstrate that our method provides significant improvements in the results compared to the current methods. We also point out some issues with the data, which may potentially change the outcome of learning-based methods.

Related Work

There are many approaches in this research area. To analyze them, we categorize these approaches into three main groups:

LiDAR-based 3D object detection

Many 3D object detection methods rely heavily on the quality of the LiDAR sensor. We characterize those LiDAR-based methods into two main approaches: point-format and voxel. With the advance of PointNet (Qi et al., 2017a, 2017b), deep learning models can efficiently learn features from unstructured point clouds data. In Qi et al. (2018), the authors created 2D proposal bounding boxes using a 2D detector. They used a PointNet-based network to learn the features from cropped point clouds regions for 3D box estimation. In Shi, Wang & Li (2019), the method directly generated high-quality 3D proposals from point clouds and leverage ROI-pooling to extract features for box refinement in the second stage.

In contrast with point-based methods, the voxel-based 3D detectors divide raw point clouds into regular grids for more efficient computation. By transforming the LiDAR point clouds into bird’s-eye-view representation and applying a single-stage detector, Yang, Luo & Urtasun (2018) achieved good performance in terms of accuracy and real-time efficiency. In Shi et al. (2020), the authors introduced a voxel set abstraction module to integrate semantic features from 3D voxels to sampled keypoints. By taking advantage of the multi-scaled features obtained from the voxel-based operation and the location information from PointNet-based set abstraction, they accomplished impressive results for 3D object detection.

Monocular 3D object detection based on representation transformation

In this category, the methods indirectly estimate 3D detection by transforming images into alternative representations before applying detection algorithms. In Wang et al. (2019), the authors generated 3D point clouds from pixel-wise depth estimations. The depth image obtained from the monocular image, known as Pseudo-LiDAR, was used to mimic the original point clouds from the LiDAR scan. Then, they passed the Pseudo-LiDAR through existing LiDAR-based 3D object detectors to obtain final results. This approach heavily depends on the quality of the depth image. Using a single image to estimate the depth image, the accuracy of the depth map is questionable. For most cases, it depends on the training data, and it is nontrivial to remove the bias in data.

In You et al. (2020), the authors took advantage of the previous work and utilized cheaper LiDAR sensors to de-bias the depth estimation and correct point clouds representation results. Wang et al. (2020) is another remarkable work on Pseudo-LiDAR. They realized that the foreground and background have different depth distributions. Therefore, they estimated the foreground and background depth using separate optimization objectives and decoders. This approach leads to some improvement in the pseudo point clouds. However, the above depth-based methods cannot leverage image information. To enhance Pseudo-LiDAR’s discriminative capability, the authors in Ma et al. (2019) proposed a multi-modal features fusion module to embed the complementary RGB cue into the generated point clouds representation.

In another work, Srivastava, Jurie & Sharma (2019) converted the perspective image to a bird’s-eye-view (BEV) image. They used Generative Adversarial Network (GAN) to perform BEV transformation as an image-to-image translation task. This work generated BEV grids directly from a single RGB image by designing a high fidelity GAN architecture and carefully curating a training mechanism, including selecting minimally noisy data for training. In another work, Roddick, Kendall & Cipolla (2019) used a different grid-based method. They mapped the 2D feature maps to bird’s-eye-view by orthographic feature transformation. They accumulated the extracted features over the projected voxel area. The voxel features were then collapsed along the vertical dimension to yield the final orthographic ground plane features. These approaches show that it is possible to imitate human perception when performing object detection on a single image without special depth sensors.

Monocular 3D object detection based on 2D detector

In Brazil & Liu (2019), the authors simultaneously estimated 2D and 3D boxes by introducing the 3D anchor box consisting of both 2D and 3D features. They also proposed a row-wise convolution and a 2D–3D optimization process to improve the orientation estimation. Follow the 3D anchor approach, the authors of Ding et al. (2020) integrated the estimated depth map into the network backbone and design a depth-guide convolution with dynamic local filters. With the new guidance, the estimation has improved significantly. Liu, Yuan & Liu (2021) leveraged the ground plane priors through a ground-aware convolution and anchors filtering pre-processing step.

Another approach in this direction is Liu, Wu & T’oth (2020), where the authors extended the CenterNet-based detector by adding depth and orientation estimation branches. They proposed a multi-step disentangling loss to handle different kinds of loss functions in 3D detection tasks. Some works, such as Jörgensen, Zach & Kahl (2019), and Li et al. (2020) used CNNs to learn the intermediate representation of a 3D object. They then optimized the estimated bounding box through a least square problem. These approaches mitigate the ill-pose problem by regress the visual features of vehicles on the image. The visual features can be eight corners instead of the spatial information (including depth, allocentric orientations). Then they used these hints to recover the 3D pose.

Some other approaches rely on an off-the-shelf 2D regional proposal network to generate 2D candidates, which can significantly reduce the search space of 3D. In Ku, Pon & Waslander (2019), the authors utilized MS-CNN (Cai et al., 2016) to extract 2D bounding boxes then generated 3D proposals and local point clouds based on the cropped features. Another work in Li et al. (2019) proposed a guidance algorithm with a 3D subnet to refine 3D bounding boxes from 2D proposals.

Methodology

Our proposed framework aims to improve the accuracy of the 3D object detection task from the monocular image. By introducing the depth adaptive convolution layer, we can leverage the prediction result of detection heads. We also present a new module that utilizes object’s pseudo-position inference for enhancing the 3D bounding box regression results. To demonstrate this idea, we describe the details of our geometric ground-guide module (GGGM), which infers the final location, orientation, and the 3D bounding boxes. This module utilizes the intermediate output of the detection network and the pseudo-position value to recover the object’s 3D pose via 2D–3D geometric transformation.

Center-based monocular 3D objects detection network

Figure 2 illustrates the overview of the proposed framework. Our detection network follows the idea of CenterNet (Zhou, Wang & Krähenbühl, 2019) architecture, which consists of a backbone for feature extraction followed by multiple detection heads. We employ a modified version of the DLA-34 (Yu et al., 2018) proposed in Zhou, Wang & Krähenbühl (2019) as the backbone of our network. Each detection head follows the design shown in Fig. 3, which includes a 3 × 3 depth adaptive convolution layer followed by a Rectified Linear Unit (ReLU) activation and a 1 × 1 fundamental convolution layer. Let I ∈ RH × W × 3 be the input image with the width of W and the height of H, our detection heads include:

Center head: The center head produces a heatmap M ∈ [0, 1](H/4) × (W/4) × c of 2D bounding box centers, where c is the number of object classes. Each value of channel ci presents the confidence score for an object of category Ci. The output heatmap is inversely transformed to the resolution of H × W × c via affine transformation to retrieve the location of 2D bounding box centers.

Keypoints head: Inspired by Li (2020), we estimate the location of 9 ordered 3D bounding box keypoints projected on the 2D image plane. Those points are the corners and center of the 3D bounding box. We consider the keypoint location as horizontal and vertical offsets from the corresponding 2D center. The keypoints head takes the feature maps to produce Pkp ∈ R(H/4) × (W/4) × 18 of coordinate offsets.

Pseudo-contact point head: This head estimates the projected location on the 2D perspective of the pseudo-contact point, which is described thoroughly in the object’s pseudo-position section. It follows the same approach mentioned in the keypoints head and produces Pco ∈ R(H/4) × (W/4) × 2 of coordinate offsets.

Orientation head: Due to the perspective transform from 3D coordinate to 2D image plane, it is impossible to regress global yaw rotation θ from a single image (see Appendix for more details). Hence we choose to regress the observation angle α by following the angle decomposition proposed in Brazil et al. (2020), which formulates the orientation into three components: heading, axis, and offset. We encode the offset angle α as [sin(α), cos(α)]T. The output of the orientation head is O ∈ R(H/4) × (W/4) × 6.

Dimension head: Dimension head directly regresses the absolute value D ∈ R(H/4) × (W/4) × 3 of object dimension. Instead of applying the regression strategy proposed in Liu, Wu & T’oth (2020) and Li (2020), which predict the value of height, width, length with a specific order, we estimate the object dimension in a more flexible way.

Figure 3 Depth adaptive convolution detection head.

The depth adaptive 2D convolution processes image features with the external guidance of generated depth maps from a pre-trained monocular depth estimation network. The output goes through a ReLU activation followed by a standard 2D convolution layer.

The visual appearance of the object has a strong impact on the object’s metrics, as illustrated in Fig. 4. Therefore, we attempt to dynamically decode the width and length of the object as follows:

(1) h=D0

(2) w={D1,if|sin(α)|>|cos(α)|D2,otherwise

(3) l={D2,if|sin(α)|>|cos(α)|D1,otherwise

where h, w, l, α are object’s height, width, length, and the observation angle of the object accordingly. The term Di is the ith channel of dimension head’s output.

Figure 4 The visual appearance of the object’s dimension with different observation angles greatly affects the estimated size of the bounding box.

Depth adaptive convolution layer

In the context of traffic scenes, vehicles usually occlude others. We observe that the detection result of a specific object should not be affected by the features of others. The irrelevant features could belong to adjacent objects or background objects. We aim to enhance the detection by selecting valuable features. Features at pixels inside the object should contribute more to 3D detection. Instead of using instance segmentation as a guide for the detection network, we use the 3D surface of the object taken from the depth map. We can always figure out the discontinuity between objects just by probing the depth map.

As a consequence, we design the Depth Adaptive Convolution Layer to handle irrelevant local structures of traditional 2D convolution. Our proposed layer injects the information from depth predictions by applying pixel-adaptive convolution from Su et al. (2019). The formulation of pixel-adaptive convolution is defined as follows:

(4) vi′=∑j∈Ω(i)K(fi,fj)W[pi−pj]vj+b

where vi′ is the convolution filtering output at pixel i, f ∈ RD are the pixel features that guide the pixel-adaptive convolution, ω(i) denotes a convolution window, pi = (xi, yi)T are pixel coordinates, W are the filter weights of convolution, v is the input features, and b are the biases value of convolution. K is a fixed kernel function.

Inspired by the work in Su et al. (2019), we use depth maps as the external guiding features for the depth adaptive convolution layer to explicitly encourage the model to extract features from pixels of corresponding objects:

(5) vi′=∑j∈Ω(i)K(di,dj)W[pi−pj]vj+b

where d ∈ RH × W × 1 is the depth estimation of the current image. K is a fixed Gaussian kernel function to calculate the correlation of guiding features:

(6) K(di,dj)=e−12(di−dj)2

Since the operator locally adapts the filter weights using the depth information, we name this operator “depth adaptive convolution”. For feature map v ∈ Rh × w × c, our proposed convolution operator generates h × w adaptive filters for each particular pixel by performing the Hadamard product of W and the local kernel K.

Losses

Our total loss comprises a heatmap loss of 2D center, regression loss for keypoints and contact point offsets, a composition loss for local orientation, a regression loss for object’s dimension, and a geometric position loss. We define this loss as follows:

(7) L=∑iλi⋅Li

where Li indicates Lheat, Lkps, Lcps, Ldim, Lori, Lpos, which are the 2D center heatmap loss, keypoint offset loss, contact point offset loss, dimension regression loss, local orientation loss and position loss, respectively. The parameters λi are [λheat, λkps, λcps, λdim, λori], which are set to [1, 1, 1, 2, 0.2] accordingly. The λpos is determined by the ramp-up function proposed in Laine & Aila (2017), which is computed as follows:

(8) λpos={0,ifne<Tmine−5(1−neTmax)2,ifTmin<ne≤Tmax11,ifne>Tmax

where ne indicates current epoch number. The ramp-up period parameters [Tmin, Tmax] are set to [40,100]. In this work, we apply focal loss in Zhou, Wang & Krähenbühl (2019) for 2D center heatmap:

(9) Lheatc=−1N∑1c∑1H/4∑1W/4{(1−M^xyc)αlog(M^xyc),ifMxyc=1(1−Mxyc)β(M^xyc)αlog(1−M^xyc),otherwise

For keypoints offsets, pseudo-contact point offsets and object’s dimension regression, we use the L1-loss. In term of observation angle regression, we employ the orientation loss function in Brazil et al. (2020).

Based on Li (2020), we can recover an object’s position by solving an over-determined system of equations using singular value decomposition, which is differentiable. Since we deduce the position’s equation system from nine keypoint offsets, orientation, and dimension, the position loss can be back-propagated through all the heads of our network. In some cases, the distance between the predicted 3D center and the groundtruth is relatively small, but the predicted bounding box is not accurate. Figure 5 illustrates this phenomenon: a little shift in angle and center can lead to significant error in 3D pose estimation. Therefore, we propose a position loss function as an L2-loss of eight corners of the predicted bounding box and the groundtruth box instead of calculating position loss using the 3D center coordinate by combining and transforming the estimated position, orientation, and dimension into a 3D bounding box. We define the position loss as follows:

(10) Lpos=18∑i=07‖Cor^i−Cori‖2

where Cori is the ith corner of the 3D bounding box.

Figure 5 Illustration of ground truth (red color) and predicted (blue color) 3D bounding box inbird’s-eye-view.

Geometric ground-guide module

Inspired by the Geometry reasoning module (GRM) from Li (2020), we introduce a 2D–3D transformation pipeline to reconstruct the object’s 3D pose, named Geometric Ground-Guide Module (GGGM). First, this module estimates a pseudo-position for every detected object. Then, it takes the outputs from the detection network comprising a 2D center, dimension, orientation, nine predicted keypoints, and the pseudo-position to generate a system of geometric equations whose solution is the position of the 3D bounding box center.

Object’s pseudo-position

We propose a lightweight approach to approximate the 3D bounding box center’s y-coordinate and z-coordinate for each detected object. Our method incorporates estimated values from the pseudo-contact point head and camera calibration to calculate the pseudo-position.

In most driving scenarios, the ground around the car is flat. This assumption is not too strong since it holds in most cases. Even if our car is on the hill, the relative position between our car and other cars should be on the same flat plane. In that case, we can apply the ground-guide module to estimate the pseudo-position.

Thus, we do not consider non-flat ground-planes, which are not available in most datasets. We assume that the principal optical axis of the camera is parallel to the ground plane. Let G be a point on the ground plane (ground point) with the corresponding 3D location (x, y, z) and pixel coordinate (u, v) on the image plane. According to the pin-hole camera model, we calculate the depth value z for each ground pixel as:

(11) z=fy⋅hcam+Tyv−cy

where fy, hcam, cy, Ty is the focal length, camera height, principal point coordinate, and relative translation, respectively.

The hcam value depends on the dataset’s camera settings, particularly 1.65 m for the KITTI dataset (Geiger, Lenz & Urtasun, 2012). Eq. (11) describes the “ground plane model”. As shown in Fig. 6, we can indicate every ground point’s depth value by knowing its vertical coordinate on the image plane. We can calibrate this extrinsic information for the camera before using it in GGGM. Figure 7 illustrates terms in the object’s pseudo-position inference process. For every object with the 3D bounding box center P at (x, y, z), we define the projection of P on the ground plane Pg at location (x, hcam, z) called pseudo-contact point. The coordinate (u, v) of Pg projected on the 2D image plane is regressed directly in the pseudo-contact point head of the detection network. Finally, the approximation of z is inferred using v and the ground plane model in Eq. (11), proclaimed as pseudo z-coordinate. We define the pseudo y-coordinate as:

(12) y=hcam−hobject2

Figure 6 The depth of the ground plane generated from the extrinsic information of the camera.

Figure 7 Object’s pseudo-position P and related terms in the inference process using camera model and ground plane model.

2D–3D transformation

With the ground model and the pseudo-position, we now derive the 3D pose reconstruction from 2D–3D geometric constraints. We enhance the 2D–3D transformation process of Li (2020) with our proposed pseudo-position.

First, the pin-hole camera model gives us a simple projection from a world point N(3d) = [x, y, z]T to a point N(2d) = [u, v]T on the image plane as follows:

(13) [xyz]=[z×u−cxfxz×v−cyfyz]

We normalize the N(2d) and denote N~(2d)=[u~,v~]T=[u−cxfx,v−cyfy]T. Thus the 3D coordinate of nine keypoints of the object can be calculated as:

(14) Kpi(3d)=zi×[Kp~i(2d)1]3×1,i=0,...,8.

On the other hand, these coordinates can be inferred from object’s attributes, including dimension D^=[l^,h^,w^]T, orientation O^=[θx,θy,θz]T and location P:

(15) Kpi(3d)=Ry(O^)×Diag(D^)×Coori+P,i=0,...,8.

where Ry(O^) is the rotation matrix around the y-axis. The angle θy can be calculated by the projected 3D center Kp^8 (obtained from the detection heads) and the observation angle α^ (See Appendix for more details).

Ry(O^)=[cos(θy^)0sin(θy^)010−sin(θy^)0cos(θy^)]3×3

Diag(D^) is the diagonal matrix of three dimensions length, height, width:

Diag(D^)=[l^000h^000w^]3×3

Coor is the matrix in which each column contains the relative coordinate of nine keypoints to the object’s center.

Coor=[1/21/2−1/2−1/21/21/2−1/2−1/201/21/21/21/2−1/2−1/2−1/2−1/201/2−1/21/2−1/21/2−1/21/2−1/20]3×9

and P = [px, py, pz]T is the position of the 3D bounding box’s center.

From (14) and (15), we deduce to these equations:

(16) zi×[Kp~i(2d)1]3×1=Ry(O^)×Diag(D^)×Coori+P,i=0,...,8

Using elementary row operations, we transform these equations to an over-determined system of 18 linear equations of variable P:

(17) [−10Kp~x0(2d)0−1Kp~y0(2d)−10Kp~x1(2d)0−1Kp~y1(2d)−10Kp~x2(2d)0−1Kp~y2(2d)−10Kp~x3(2d)0−1Kp~y3(2d)−10Kp~x4(2d)0−1Kp~y4(2d)−10Kp~x5(2d)0−1Kp~y5(2d)−10Kp~x6(2d)0−1Kp~y6(2d)−10Kp~x7(2d)0−1Kp~y7(2d)−10Kp~x8(2d)0−1Kp~y8(2d)]18×3[pxpypz]3×1=[cos(r^y)l^2+sin(r^y)w^2−Kp~x0(2d)(−sin(r^y)l^2+cos(r^y)w^2)h^2−Kp~y0(2d)(−sin(r^y)l^2+cos(r^y)w^2)cos(r^y)l^2−sin(r^y)w^2−Kp~x1(2d)(−sin(r^y)l^2−cos(r^y)w^2)h^2−Kp~y1(2d)(−sin(r^y)l^2−cos(r^y)w^2)−cos(r^y)l^2−sin(r^y)w^2−Kp~x2(2d)(−sin(r^y)l^2−cos(r^y)w^2)h^2−Kp~y2(2d)(sin(r^y)l^2−cos(r^y)w^2)−cos(r^y)l^2+sin(r^y)w^2−Kp~x3(2d)(sin(r^y)l^2+cos(r^y)w^2)h^2−Kp~y3(2d)(sin(r^y)l^2+cos(r^y)w^2)cos(r^y)l^2+sin(r^y)w^2−Kp~x4(2d)(−sin(r^y)l^2+cos(r^y)w^2)−h^2−Kp~y4(2d)(−sin(r^y)l^2+cos(r^y)w^2)cos(r^y)l^2−sin(r^y)w^2−Kp~x5(2d)(−sin(r^y)l^2−cos(r^y)w^2)−h^2−Kp~y5(2d)(−sin(r^y)l^2−cos(r^y)w^2)−cos(r^y)l^2−sin(r^y)w^2−Kp~x6(2d)(sin(r^y)l^2−cos(r^y)w^2)−h^2−Kp~y6(2d)(sin(r^y)l^2−cos(r^y)w^2)−cos(r^y)l^2+sin(r^y)w^2−Kp~x7(2d)(sin(r^y)l^2+cos(r^y)w^2)−h^2−Kp~y7(2d)(sin(r^y)l^2+cos(r^y)w^2)00]18×1

where each keypoint is associated with two geometric equations. The system of Eq. (17) is the baseline transformation in Li (2020). The over-determined system of linear equations AP = b is solved by using the ordinary least square method. The intuition is minimizing the least square cost function:

(18) e(P)=‖b−AP‖2

then the approximate solution P=argminP⁡‖b−AP‖2=(ATA)−1ATb.

In practice, objects at divergent locations have different regression error which mostly comes from the inaccurate regressed keypoints’ offsets. Despite the fact that there is no difference in 3D dimensions among distant and near objects, near objects have bigger 2D sizes than the far-away objects in 2D images. Consequently, a little shift of keypoints’ offsets of a distant object can lead to a significant 3D pose error while there is a small change in the 3D pose in the case of near object. To accommodate for this phenomenon, we introduce a L2 regularization term inferred from the pseudo-position and add it into the cost function (18):

(19) e(P)=‖b−AP‖2+Λ‖P−Ppseudo‖2

where λ and Ppseudo are pre-defined scale factor and an initial value for the position P (yps and zps are the pseudo y and z-coordinate) accordingly. The L2 regularization term encourages the solution P to not only depend on the 2D regressed output but also satisfy the ground plane model (illustrated in Fig. 6).

The initial position Ppseudo follows the ground plane model that we proposed before. The final position P is computed as:

(20) P=(ATA+Λ)−1(ATb+ΛPpseudo)

We design a soft scaling factor λ in Eq. (19) adaptive to the location of objects. In particular, a distant object has bigger λ, meaning that the refined 3D pose will depend less on the accuracy of keypoints’ offsets regression then the error can be reduced. Distance of object is roughly estimated by the y-position of the 2D bounding box, the smaller y-position is, the farther object locates. Nevertheless, the two components of Ppseudo do not share a similar scale. While the y-axis location only varies within meters, the z-position ranges from 0 up to hundreds of meters. Therefore, we choose the scaling factor λ to ensure that neither the y-component term nor z-component term dominates the other. The formula of the scale factor λ = [λ x, λ y, λ z]T is as follows:

(21) {λx=0λy=0.5e−y2d−yminymax−yminλz=0.0025λy

where y2d is the y-position of the 2D bounding box, ymin and ymax are set to 170 and 384.

The refined position depends on both the pseudo-position and 2D–3D constraints. Thus, even if two cars are not on the same plane, the error of the obtained pseudo-position does not affect much on the refined position. If the assumption holds, the pseudo-position can enhance the final pose estimation, as we showed in the experiments. Otherwise, a good pose obtained from the 2D regression network can produce an acceptable result since the pseudo-position won’t affect much in this case. We designed this regularization term as a soft regularization scheme with the scaling factor based on the y-position of the object on the 2D image plane. With this adaptation, our approach can improve the pose estimation result in various conditions.

Experiments

Dataset and setting

The object detection task of the KITTI dataset (Geiger, Lenz & Urtasun, 2012) contains a total of 7,481 training images and 7,518 test images. Because of the lack of labels in the test set, we follow Chen et al. (2015) to split the training set into 3,712 samples for training and 3,769 samples for evaluation purpose. We train our model on the trainval set and evaluate the result on the test set to compare with the other methods while training on the train set and testing on the val set for ablation study.

Implementation details

We implemented our model using Pytorch 1.7, CUDA 10.2, CuDNN 7.5.0 on Ubuntu 18.04 on a machine with a single RTX2080. The input image is normalized to the resolution 1,280 × 384. We also perform the data augmentation process, including color jittering, horizontal flipping, random scaling, and random shifting, using the default setting of Zhou, Wang & Krähenbühl (2019) with the chances of 100%, 50%, and 70%, respectively. Because scaling and shifting are 3D-coordinate inconsistent, we formulate these translations as a single affine transformation. Therefore the model’s output can be converted into the original coordinate of the input image, make the data augmentation independent of the coordinate. We use our depth adaptive convolution layer for every detection head. Bhat, Alhashim & Wonka (2020) is the pre-trained monocular depth estimation network for depth guidance. The depth maps are scaled down with the factor of 14 by nearest-neighbor interpolation before passing through the depth adaptive convolution layer.

We employ the Adam optimizer and use the base learning rate at 10−4 for the training process. This learning rate is scheduled to decrease by the factor of 10 at epoch 40 and 90. For efficiency, the batch size we select is not too large and is an adequate size. During the pose estimation process, we apply a 3 × 3 max-pooling operator on the head’s output and pick the top 40 objects based on the 2D confidence scores. We select the threshold for this score to be 0.3, and there is no need for applying Non-Maximum Suppression (NMS) in the testing phase.

Comparative results

The KITTI benchmark for the 3D object detection task consists of 2 principal metrics: average precision for 3D intersection-over-union (AP|3D) and average precision for bird’s-eye-view (AP|BEV), which are separated into three difficulty levels Easy, Moderate, and Hard according to the bounding box’s height, occlusion, and truncation level (Geiger, Lenz & Urtasun, 2012). From October 2019, following Simonelli et al. (2020), the evaluation metrics are changed from 11-point Interpolated Average Precision (AP) metric AP|R11 to 40 recall positions-based metric AP|R40. By default, the KITTI benchmark requires 3D bounding boxes with the Intersection over Union (IoU) of 70% for the Car category. We report our results with 11 other recent methods ordered by the AP |R403D of the Moderate difficulty level of the Car category. As observed from Table 1, our method achieves remarkable improvement in comparison to contemporary monocular 3D object detection frameworks. It is notable that our proposed approach outperforms all existing approaches in the Easy and Mod difficulty levels on the test set. Comparing with the second-best competitor, we achieve 17.75 (↑ 1.02) for Easy and 12.00 (↑ 0.28) for Moderate.

Table 1 Comparative results on the KITTI 3D object detection test set of the Car category.

Best results are marked in bold.

Method	Backbone	AP |R403D (IoU = 0.7)	AP |R40BEV (IoU = 0.7)	
		Easy	Mod	Hard	Easy	Mod	Hard	
ROI-10D (Manhardt, Kehl & Gaidon, 2019)	ResNet-34	4.32	2.02	1.46	9.78	4.91	3.74	
GS3D (Li et al., 2019)	VGG-16	4.47	2.90	2.47	8.41	6.08	4.94	
MonoPSR (Ku, Pon & Waslander, 2019)	ResNet-101	10.76	7.25	5.85	18.33	12.58	9.91	
M3D-RPN (Brazil & Liu, 2019)	DenseNet-121	14.76	9.71	7.42	21.02	13.67	10.23	
SMOKE (Liu, Wu & T’oth, 2020)	DLA-34	14.03	9.76	7.84	20.83	14.49	12.75	
MonoPair (Chen et al., 2020)	DLA-34	13.04	9.99	8.65	19.28	14.83	12.89	
RTM3D (Li et al., 2020)	DLA-34	14.41	10.34	8.77	19.17	14.20	11.99	
AM3D (Ma et al., 2019)	ResNet-34	16.50	10.74	9.52	25.03	17.32	14.91	
PatchNet (Ma et al., 2020)	PointNet-18	15.68	11.12	10.17	22.97	16.86	14.97	
KM3D (Li, 2020)	DLA-34	16.73	11.45	9.92	23.44	16.20	14.47	
D4LCN (Ding et al., 2020)	ResNet-50	16.65	11.72	9.51	22.51	16.02	12.55	
Ours	DLA-34	17.75	12.00	9.15	25.80	16.93	12.50	

That said, our estimation results on the Hard test set seem better. There are some abnormal detection cases in the KITTI dataset that significantly show the robustness of our proposed method. Figure 8 shows some particular cases where our method produces better detection results. We show more experiments on these abnormal cases in the Appendix.

Figure 8 Illustration of unlabeled cases in KITTI val set.

First row: monocular image. Second row: KITTI’s ground truth. Third row: the results from Li (2020). Last row: our prediction results.

Let’s take a closer look at the particular sample we illustrate in Fig. 8. In the figure, we mark some points with numbers where the difference in the estimation occurs. At point #1, we can see an occluded car in the original image. This car is completely unlabeled on the groundtruth, while both KM3D Li (2020) and our method can detect. The size of the car at that position is a bit different due to the ground assumption we posed earlier. In case the bottom of the object is occluded, our assumption can lead to better estimation. At point #2, there are actually two cars packing nearby each other and the white one is heavily occluded by the black one. In this case, the groundtruth of KITTI only marks an object in the middle of two cars (a bit bias to the black bar). The result of KM3D Li (2020) is similar to the one of groundtruth, while our detector can detect both cars. After visualizing the result, we verified that the detected location of the two cars was not overlapped and consistent with the context of the input image. The position #3 also demonstrates another case where the KITTI groundtruth ignores the object while both KM3D Li (2020) and our method can detect. Likewise, in the case of position #4, the visible car is completely unlabeled in the groundtruth, while both KM3D Li (2020) and ours can detect it. If we observe carefully, the detection box of ours fit on the whole car (very close to the groundtruth at position #5 too). This observation explains why the score of our method is not as good as other methods in the Hard test set. The robustness labels in the groundtruth play an important role in the evaluation result.

We show qualitative results in Fig. 9. The results from the left column are inferred from the val set to compare our predictions with groundtruth labels. The right column images are taken from the official test set.

Figure 9 Qualitative illustration of our monocular 3D detection results (left: val set, right: test set).

Green color: our predictions; red color: ground truth; dot: projected 3D center; diagonal cross: heading of object.

Ablation Study

Accumulated impact of our proposed methods

Table 2 shows the experimental results that we conduct to measure the impact of each component on our proposed designs. In these experiments, we show the contribution of each proposed component to the overall performance of the monocular 3D object detection task. We follow the default setup of Li (2020) to train the baseline model. We use the 40 recall positions Average Precision (AP|R40) metric to evaluate the following experimental results for a fair comparison with the official KITTI benchmark.

Table 2 Evaluation on accumulated improvement of our proposed methods on KITTI val set.

“P.” denotes pseudo-position, “DA.” denotes depth adaptive convolution. Best results are marked in bold.

Method	AP |R403D (IoU = 0.7)	AP |R40BEV (IoU = 0.7)	
	Easy	Mod	Hard	Easy	Mod	Hard	
Baseline	11.56	10.31	8.94	19.86	16.33	14.56	
+DA.	15.12	12.02	10.84	23.13	18.24	15.84	
+P.	16.15	13.17	11.48	25.17	19.91	17.63	
+P. +DA.	17.59	14.79	13.10	25.01	20.56	18.20	

Evaluation on depth adaptive convolution

We analyze the impact of the proposed depth adaptive convolution on training phase convergence by comparing training losses between standard convolution and depth adaptive convolution detection heads. From Fig. 10, we observe that the depth adaptive convolution detection heads yield more agile training convergence and better adaptation in the training phase than the original 2D convolution heads. Especially in terms of the supervised loss of keypoints head, our approach significantly improves training loss’s stability.

Figure 10 Trajectories of the optimization process for each detection head with standard convolution and depth adaptive convolution operation.

Evaluation on geometric ground-guide module

For demonstrating the impact of the Geometric Ground-Guide Module, we evaluate four alternatives: the model with Depth Adaptive Convolution (+DA.) and without pseudo-position refinement, the +DA. model with pseudo y-coordinate, the +DA. model with pseudo z-coordinate, and the +DA. model with full pseudo-position. The result in Table 3 shows that with only pseudo-position in y-coordinate or z-coordinate gives a significant improvement for all metrics. Especially when combining these y and z-component into our proposed pseudo-position, the detection’s result can be improved by more than 23%.

Table 3 Impact of geometric ground-guide module for 3D object detection on the KITTI val set.

“DA.” denotes the model using depth adaptive convolution. Best results are marked in bold.

Method	AP |R403D (IoU = 0.7)	AP |R40BEV (IoU = 0.7)	
	Easy	Mod	Hard	Easy	Mod	Hard	
+DA	15.12	12.02	10.84	23.13	18.24	15.84	
+DA + pseudo y-coordinate	16.22	12.51	11.24	24.76	18.81	16.33	
+DA + pseudo z-coordinate	16.92	13.28	11.38	24.97	18.96	16.40	
+DA + pseudo y and z-coordinate	17.59	14.79	13.10	25.01	20.56	18.20	

Evaluation on the impact of depth estimation quality on adaptive convolution

In this experiment, we analyze the performance of the depth adaptive convolution with different pretrained depth models. To study the impact of depth estimation quality on our 3D detection results, we generate different depth maps from three recent supervised monocular depth estimation methods: DORN (Fu et al., 2018), BTS (Lee et al., 2020), and AdaBins (Bhat, Alhashim & Wonka, 2020). Then, we apply the model using the depth adaptive convolution with pseudo-position refinement to evaluate the results on the KITTI val set. As shown in Table 4, the model with depth maps estimated from Bhat, Alhashim & Wonka, 2020-the current state of the art among monocular depth estimation methods, obtains the highest performance. Besides, as depth guidance only provides a better geometric structure for dense traffic scenes, the accuracy on the Easy level of the 3D object detection task does not greatly depend on the depth estimation quality. This is because in the Easy dataset, we do not have small and occluded objects.

Table 4 Comparisons of different depth estimation quality for 3D object detection on KITTI val set.

We use an asterisk (*) to indicate using standard convolution operation for our detection network. Best results are marked in bold.

Depth estimator	AP |R403D (IoU = 0.7)	AP |R40BEV (IoU = 0.7)	
	Easy	Mod	Hard	Easy	Mod	Hard	
None*	16.15	13.17	11.48	25.17	19.91	17.63	
Dorn	17.44	13.85	12.52	25.86	20.37	17.95	
BTS	17.57	14.21	13.06	25.00	20.54	18.19	
AdaBins	17.59	14.79	13.10	25.01	20.56	18.20	

Conclusion

In this work, we propose a novel framework for monocular 3D object detection. Consolidating the strength of the convolutional neural network and geometric constraints, our proposed approach aims to compensate for the lack of depth information in the monocular image. We introduce a novel convolution layer, named Depth Adaptive Convolution, to improve the accuracy and stability of the detection network. We achieve this by leveraging the guidance from a pre-trained monocular depth estimator. We then propose the Geometric Ground-Guide Module, which takes advantage of 2D–3D geometric information and constraints in driving scenarios, to accurately recover the object’s 3D pose from 2D regression results.

As demonstrated in the experiments, our proposed approach yields better performance and outperforms many current state-of-the-art methods. We also perform analysis on the Hard cases where we point out that the groundtruth sometimes is not entirely correct, which leads to some differences in the qualitative results. Therefore, the evaluation numbers do not reflect the performance of estimation methods.

In the future work, we will conduct experiments on different datasets such as Waymo, CityScapes as well as the data from the traffic scenarios in our country to demonstrate the performance of our proposed method.

Appendix

Egocentric and allocentric orientation in 3D coordinate

In Fig. 11, we illustrate the difference between the egocentric and allocentric angles in the bird’s-eye-view. The egocentric angles of the two cars change in respect to the camera’s viewpoint, while their allocentric angles remain the same. The allocentric angle (θ) can be deduced from the egocentric angle (α) and the ray angle (ray), the angle between the z-axis and the ray passing through the object’s 3D center.

Figure 11 Egocentric (green color) and allocentric (orange color) angles in the bird’s-eye-view.

Red arrow indicates the heading of the car, while blue arrow is the ray between the origin and the car’s center.

(22) θ=α+ray

If C = (u, v) is the project 3D center of object on image plane, the Eq. (22) can be rewritten as:

(23) θ=α+arctanu−cxfx

where fx and cx are the focal length and principal point coordinate of the camera.

Visualization of the impact of pseudo-position

We note here one example to show the impact of the pseudo-position on the detection result. Figure 12 illustrates a common case on the road. We show the bird’s-eye-view of the detection result with and without the help of the pseudo-position. One can easily see that the detection result matches with the groundtruth when we employ the pseudo-position.

Figure 12 Visualization of the impact of pseudo-position for refining object’s positions.

(A) Input image. (B & C) The 3D detection results without and with the pseudo-position refinement, respectively. Red is ground truth z-position, green is the predicted z-position.

Abnormal detection cases of the KITTI dataset

In Fig. 13, we plot the groundtruth labels (red boxes) and our predictions (green boxes) of additional abnormal cases where the vehicles are not well-labeled.

Figure 13 Abnormal detection cases from KITTI val set.

Left: ground truth labels, right: our predictions.

We can observe similar phenomenons like what we have seen in Fig. 8. While investigating the dataset, we find out that the dataset contains many data samples like these, leading to some differences in the benchmark. While the KITTI benchmark is a good standard to measure the performance of detection methods, finding abnormal detection raises a concern on the robustness of detection results. Perhaps, some methods may overfit the dataset or have been finetuned to match with such cases.

Occlusion is something that we encounter a lot in practice. If we ignore the occlusion cases, like what we have seen in these experiments, it would be dangerous, especially in autonomous driving scenarios.

Supplemental Information

Supplemental Information 1 Code.

Click here for additional data file.

Additional Information and Declarations

Competing Interests

Author Contributions

Data Availability

The authors declare that they have no competing interests.

Minh-Quan Viet Bui conceived and designed the experiments, performed the experiments, analyzed the data, performed the computation work, prepared figures and/or tables, authored or reviewed drafts of the paper, and approved the final draft.

Duc Tuan Ngo conceived and designed the experiments, performed the experiments, analyzed the data, performed the computation work, prepared figures and/or tables, authored or reviewed drafts of the paper, and approved the final draft.

Hoang-Anh Pham analyzed the data, authored or reviewed drafts of the paper, and approved the final draft.

Duc Dung Nguyen conceived and designed the experiments, analyzed the data, authored or reviewed drafts of the paper, and approved the final draft.

The following information was supplied regarding data availability:

The code is available as a Supplemental File.

The KITTI dataset is available from http://www.cvlibs.net/datasets/kitti/.

The LaTeX files used during the review process are available at figshare: Nguyen, Duc Dung (2021): Experiment-ground plane model. figshare. Figure. https://doi.org/10.6084/m9.figshare.15000432.v1

Nguyen, Duc Dung (2021): peerj_arch.jpg. figshare. Figure. https://doi.org/10.6084/m9.figshare.15000429.v1

Nguyen, Duc Dung (2021): KITTI Submission. figshare. Figure. https://doi.org/10.6084/m9.figshare.14452614.v1

Nguyen, Duc Dung (2021): Evaluation Tables. figshare. Figure. https://doi.org/10.6084/m9.figshare.14452596.v1

Nguyen, Duc Dung (2021): Abnormal Detection Cases. figshare. Figure. https://doi.org/10.6084/m9.figshare.14452593.v1

Nguyen, Duc Dung (2021): Qualitative Results. figshare. Figure. https://doi.org/10.6084/m9.figshare.14452608.v1

Nguyen, Duc Dung (2021): Figures. figshare. Figure. https://doi.org/10.6084/m9.figshare.14452611.v1

Additional results are available from The KITTI Vision Benchmark Suite: http://www.cvlibs.net/datasets/kitti/eval_object_detail.php?&result=d869f813e45c1f5050cc5ca8a6014c89a7f85116.

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
