# Peer review of "GAC3D: improving monocular 3D object detection with ground-guide model and adaptive convolution"

_PeerJ Computer Science, doi:10.7717/peerj-cs.686_

## Round 0.1 · original submission · Major Revisions

Three mixed reviews have been received for the paper. Some concerns regarding the methodology and clarity have been flagged. Please provide detailed one-to-one responses.

Reviewer 1 ·

Basic reporting

I commend the authors for their work. The manuscript is written in unambiguous language with clearly labeled tables and figures. The authors did a good job of summarizing the literature. The bottleneck in the existing KITTI benchmark discussed in the article is an intriguing direction for the research community.

Experimental design

Monocular 3D Object detection has very significant applications in the context of autonomous navigation, and, thus it aligns within the scope of the journal. The manuscript describes the proposed approach in a detailed manner with the mathematical formulation backing it up. The provided implementation details and supplementary material should be enough to replicate the results.

Validity of the findings

The authors provide a comprehensive comparison of the proposed method with the literature. The ablation study also highlights the significance of the different parts of the method.

Reviewer 2 ·

Basic reporting

This article put forward an approach to improve existing 3D object detection from monocular images by using depth adaptive convolution (DAP) layer and ground-guide model (GGM). The authors proposed a type of convolutional neural network incorporating the DAP and GGM.
The proposed DAP is an algorithm that leverage on the method using pixel-adaptive convolution by Su et al (2019), and then work it into the context of depth related variables for each pixel. The authors stated that the DAP objective is introduced to handle irrelevant local structures of traditional 2D convolution. However, I would like to ask the authors to elaborate more on the irrelevant local structure, in which how this DAP overcomes the existing problem and how significant it is.
The proposed GGM make use of estimating the “pseudo-position”, assume that the ground plane is flat. And then this pseudo-position is put together with output from the detection network (2D centre, orientation, keypoints, …etc) to form the matrix for regression. My comments are:
(1) It may be okay to make the assumes of flat ground at this moment, however, the authors should explain a bit on could this approach be improved in the future to handle e.g. planar surfaces that are not necessary flat, from flat to uphill etc.
(2) On the scaling factor of equation (18), please describe the actual sets of values that you have used and with regards to those that yield better results.

The experiment is carried using the KITTI dataset, which is commonly used in the community for algorithm testing and evaluation. The authors reported that their method yield better results in most situations as compared to existing approaches.

Others:
(1) "Cori" in equation (9) not explained.
(2) The second and third contribution as stated in pg3-4 seem to be fairly inter-related, therefore it would be better if you could combine them i.e. mention the GGM together with “pseudo-position” estimation.
(3) Suggest to reword and rephrase last sentence in pg20: “Let’s put aside non-flat …” to “Thus, we do not consider non-flat ….”.

I would recommend this article to be accepted upon make the minor corrections to address all the questions that are described.

Experimental design

The experimental design is valid and sufficient comparing with existing practice. However, it would be better that in future work, the research work should capture their own data and focus more on the challenging scenarios.

Validity of the findings

The findings are valid with respect to its concept formulation and testing approach.

Additional comments

I would suggest that in the title and abstract to mention depth adaptive convolution first before ground-guided model, so that they follow the description flow of the two methods, e.g. for the title: “ … 3D object detection with depth adaptive convolution and ground-guide model”.

Reviewer 3 ·

Basic reporting

Please see the general comments below.

Experimental design

Please see the general comments below.

Validity of the findings

Please see the general comments below.

Additional comments

The paper claims the contributions as follows:
1. novel depth adaptive convolution
2. pseudo-position and ground-guide module

However, there are some critical issues with the proposed method:

- The proposed depth adaptive convolution seems to be directly adapted from Su et al. However, in Su's paper, f_i, f_j represents the feature vector at a certain pixel. However, in this paper, the feature is changed to depthmap with only one single value at a certain pixel location. Then, Eq (5) should not be simply copied from the original paper, where f_i here should be a number instead of a vector.

- The intuition behind this depth adaptive convolution is not clearly stated. Taking the depth information into the convolution operation (using Eq (5)), how can that be a better way to handle "irrelevant local structure"?

- I doubt the innovation of the pseudo-position and ground-guide module.
- First, the assumption of "ground is flat" is too strong. Even in KITTI, you cannot assume the ground plane is relatively static to the camera, which means the camera poses should be always changing while driving. Therefore, the mapping in Fig. 6 should not be a unique one for all frames.
- Second, the 2D-3D transformation is just borrowed from Li, so that should not be clearly stated with proper citation.

- The experimental results are not well-organized.
For example,
1) the author should mention the IOU criterion of the results in Table 1. Other papers usually show both IOU>0.7 and IOU>0.5.
2) The baseline of Table 2 is using the default setup in Li's paper but the results are much lower than Li's paper. Why?
3) The baseline in Table 2 and Table 3 should not be the same for a fair comparison. The baseline in Table 3 should include DA.
4) Table 4 is not clear. What is the purpose to show three others depth estimation results and compare them with standard convolution? I think the proper way to show the depth estimation results should be by comparing your depth estimation results with other SOTA methods.

---

## Round 0.2 · accepted · Accept

The reviewers did not respond to this version. I checked the revised version and the response letter. All that needs to be patched has been patched. Therefore, I would like to recommend an acceptance. Cheers.